# Response to photodynamic therapy combined with intravitreal aflibercept for polypoidal choroidal vasculopathy depending on fellow-eye condition:2-year results

Mio Matsubara[1], Yoichi Sakurada[1] *, Atsushi Sugiyama[1], Yoshiko Fukuda[1],
Ravi Parikh[2,3], Kenji Kashiwagi[1]

1 Departments of Ophthalmology, Faculty of Medicine, University of Yamanashi, Kofu, Yamanashi, Japan,
2 New York University School of Medicine, New York, NY, United States of America, 3 Manhattan Retina and
Eye Consultants, New York, NY, United States of America

* sakurada@yamanashi.ac.jp

doi.org/10.1371/journal.pone.0237330

**Editor:** Manuel Alberto de Almeida e Sousa Falcão,
Faculty of Medicine of the University of Porto,
PORTUGAL

## Abstract

We investigated whether response to photodynamic therapy (PDT) with intravitreal afliber-
cept injection (IAI) for polypoidal choroidal vasculopathy (PCV) differs depending on fellow
eye condition. A retrospective review was conducted for consecutive 60 eyes with PCV
treated with PDT combined with IAI as well as 2-years of follow-up data. Fellow eyes were
divided into 4 groups; Group 0: no drusen, Group 1; pachydrusen, Group 2; soft drusen,
Group 3: PCV/fibrovascular scarring. Best-corrected visual acuity improved at 24-months
irrespective of groups and there were no significant differences in visual improvement
among treated eyes among the 4 groups. Within 2-years, 35 (58.3%) required the retreat-
ment. The need for retreatment including additional injection and the combination therapy
was significantly less in Group 1(12.5%) compared to the others (P = 0.0038) and mean
number of additional IAI was also less in Group 1 compared to the others (P = 0.017). The
retreatment-free period from the initial combination therapy was longest in Group 1 (23.6
±1.1 months) (P = 0.0055, Group 0: 19.1±6.9, Group 2: 12.8±7.9, Group 3: 11.5±9.9). The
need for retreatment was significantly different according to fellow-eye condition. Among
PCV patients, pachydrusen in fellow eyes appear to be a predictive characteristic for a
decreased treatment burden at 2 years.

## Introduction

Polypoidal choroidal vasculopathy (PCV) is widely considered to be a unique subtype of exu-
dative age-related macular degeneration (AMD) characterized by aneurysmal dilations with
branching vascular network on indocyanine green angiography (ICGA). [1, 2] To date, PCV
has been considered to be a variant of type1 choroidal neovascularization secondary to neovas-
cular AMD and they share clinical and genetic background. [3–7] Its prevalence accounts for a

**Data Availability Statement:** All relevant data are within the manuscript and its Supporting Information files.

**Funding:** This work was supported by Japan Society for the Promotion of Science KAKENHI Grant Number 23791972 (YS). The funder provided support in the form of salaries for authors but did not have any additional role in the study design, data collection and analysis, decision to publish, or preparation of the manuscript. Manhattan Retina and Eye Consultants has provided a salary for R.P. and did not have any additional role in the study design, data collection and analysis, decision to publish, or preparation of the manuscript. The specific roles of these authors are articulated in the 'author contributions' section.

**Competing interests:** Competing interest Manhattan Retina and Eye Consultants has provided a salary for R.P. This does not alter our adherence to PLOS ONE policies on sharing data and materials. There are no patents, products in development or marketed products associated with this research to declare.

half of advanced AMD on Japanese clinic studies, although some debate that PCV is a distinct entity. [8, 9]

Photodynamic therapy (PDT) combined with intravitreal anti-vascular endothelial growth factor (VEGF) injection are a first-line treatment option for PCV along with intravitreal anti-VEGF injection monotherapy in the real-world. [10–14] PDT induces occlusion of polypoidal lesions by vaso-constrictive effect on choroid; however, it causes ischemia in the choroid and RPE, which may result in VEGF upregulation. Anti-VEGF agents reduce the up-regulated VEGF, and thus decreasing subretinal and sub-RPE exudation. Therefore, combination therapy involving PDT and intravitreal injection of anti-VEGF agents might be an ideal therapy for PCV in terms of visual improvement and occlusion of polypoidal lesion. [15, 16]

Pachydrusen are a new clinical entity characterized by isolated or scattered yellow-whitish drusenoid deposits as seen in the posterior pole of the retina over areas of a thickened choroid. [17, 18] Recent studies demonstrated that morphology under pachydrusen showed increased Haller's layer with attenuation of choriocapillaris. [19, 20] Pachydrusen are not specific to AMD and they are often seen in pachychoroid diseases such as central serous chorioretinopathy. [20] Several studies reported that pachydrusen were more prevalent in eyes with PCV compared to those with other exudative AMD. [21–23] Recently we reported clinical and genetic characteristics of pachydrusen in patients with exudative AMD. [23] However, there have been no reports investigating the treatment outcomes in eyes with PCV and pachydrusen.

In the present study, we classified eyes with PCV into 4 groups depending on untreated fellow-eye condition; Group 0; no drusen, Group 1; pachydrusen, Group 2; soft drusen, and Group 3; PCV/fibrovascular scarring and investigated 2-year results of the combination therapy for PCV and compared 2-year results among the 4 groups.

## Methods

A retrospective medical chart review was conducted for consecutive 60 patients with PCV who were initially treated with PDT combined with intravitreal aflibercept injection (IAI) between January 2013 and August 2017 and completed 2-years follow-up. This retrospective study was approved by the institutional review board of University of Yamanashi and followed the tenets of declaration of Helsinki. Written informed consent was obtained from each patient before the treatment.

Prior to the treatment, all study patients received comprehensive examination including best-corrected visual acuity measurement using Landolt chart, slit-lamp examination with 78 diopter lens, color fundus photography, fluorescein and indocyanine green angiography(FA/ICGA) (HRA-2, Heidelberg Engineering, Heidelberg, Germany), optical coherence tomography(OCT) using Spectralis (ver5.4 HRA+OCT) and/or DRI-OCT1 Atlantis(Topcon Corp, Tokyo, Japan). All OCT images were obtained by a horizontal or vertical line through the fovea. In patients with extrafoveal lesions, we performed OCT scans corresponding to the lesion in addition to scans of the fovea. Late phase images of FA/ICGA were obtained 10 minutes after the dye injection. Using OCT sans, soft drusen were identified RPE elevations while pachydrusen were sub-retinal piment epithelial deposits. Central retinal thickness (CRT) was measured as the vertical distance between inner surface of neurosensory retina and retinal pigment epithelium beneath the fovea. Subfoveal choroidal thickness was measured as the vertical distance between Bruch's membrane and choroidoscleral border at the fovea.

PCV was diagnosed as we previously described. [9] All PCV cases showed solitary or multiple aneurysmal dilations (polypoidal lesions) with or without branching vascular networks on

ICGA and irregular retinal pigment epithelium (RPE) elevation with serous/hemorrhagic detachment of neurosensory retina and/or RPE on OCT.

Choroidal vascular hyperpermeability (CVH) was defined as multifocal hyperfluorescent area with blurred margins within the choroid that increased the intensity in the late phase ICGA as we previously described. [9]

### Treatment

Intravitreal injection of aflibercept (2.0mg/0.05ml) (Bayer AG, Leverkusen, Germany) was administrated for all study eyes 1 week before PDT. PDT was performed according to a standard protocol: verteporfin (Visudyne, Novartis) was administrated intravenously(6mg/mm$^2$) for 10 minutes. Fifteen minutes after the verteporfin injection, a pulse of 689-nm- wavelength light was delivered using a diode laser unit (Visulas PDT system 690S, Carl Zeiss) for 83 seconds with an intensity of 600mW/cm$^2$. Greatest linear dimension (GLD) was defined as covering all areas including branching vascular networks and polypoidal lesions on ICGA. Spot size was defined as 1000μm in addition to GLD.

### Follow-up and retreatment

After the initial combination therapy all study patients were followed every 3 months until recurrence. Recurrence was defined as exudative changes including subretinal fluid or hemorrhage as seen on OCT or subretinal/sub-RPE hemorrhage seen on ophthalmoscopy. When recurrence was seen, additional FA/ICGA was performed. Additional combination therapy was administrated when a polypoidal lesion with or without branching vascular networks was seen on ICGA. Additional IAI was administrated when branching vascular networks without polypoidal lesions were seen on ICGA. After first recurrence all patients were followed-up every month.

### Classification of groups

Depending on untreated fellow eye conditions, patients were subdivided into 4 groups: Group 0 (no drusen), Group 1 (pachydrusen), Group 2 (soft drusen), Group 3 (PCV/scarring). SD-OCT and late phase ICGA was used to differentiate pachydrusen from soft drusen. Soft drusen exhibited hypofluorescent on late phase ICGA, [24, 25] on the other hand pachydrusen exhibited hyperfluorescent on late phase ICGA. [17, 23] Presence or absence of drusen was judged in the 45°color fundus photography of the untreated fellow eye. Group classification was independently performed by 2 graders (M.M, and Y.S). Discordant diagnosis was resolved through open arbitration. A representative case with pachydrusen was shown in Fig 1 and a representative case with soft drusen was shown in Fig 2.

### Statistical analysis

Statistical analysis was performed using DR. SPSS. Differences of continuous variable and categorical variables were analyzed by Mann-Whitney U test and chi-square test, respectively. Differences of values between before and after treatment were analyzed by Wilcoxon signed rank test. Log-rank test was used to compared retreatment-free period among the 4 groups. P-value less than 0.05 was considered statistically significant.

## Results

Table 1 shows demographic characteristics of patients with PCV in each group. Group 2(fellow eyes with soft drusen) is significantly older compared with other groups (Group 0:

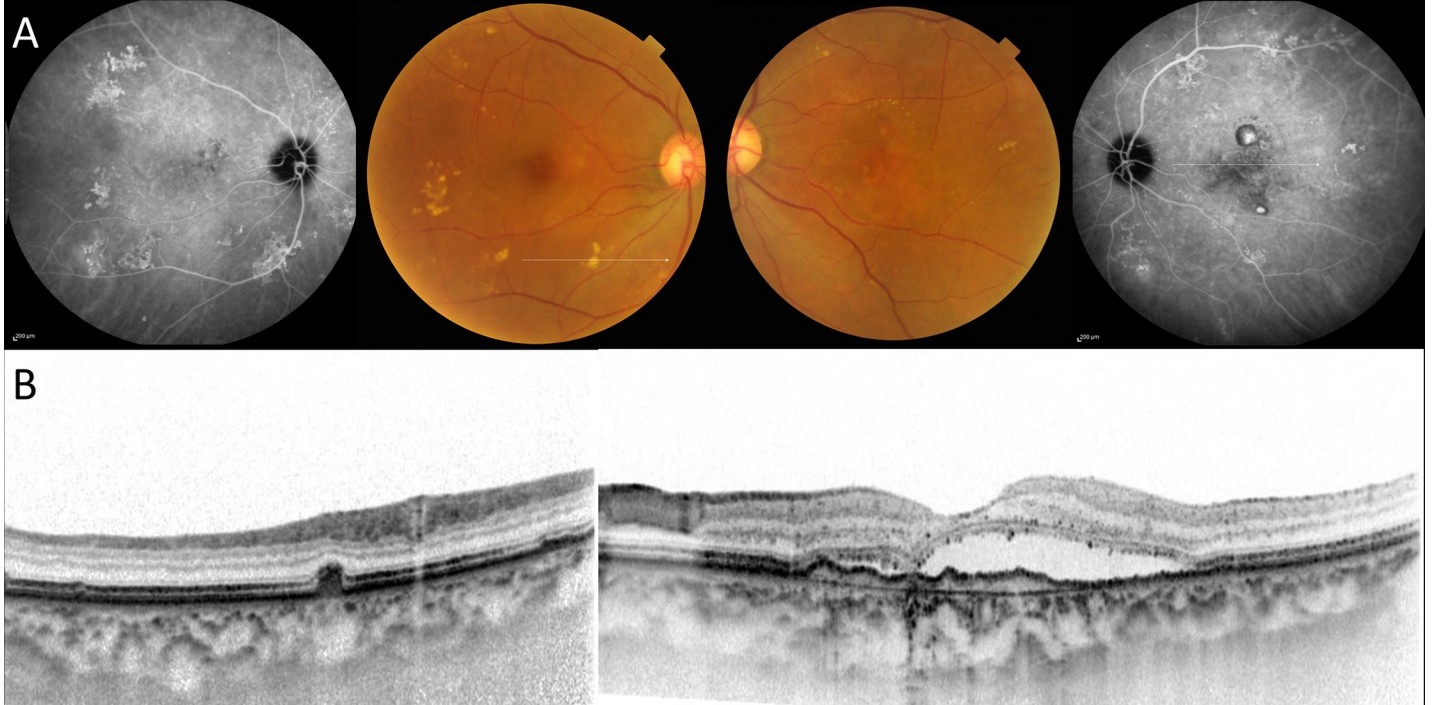

**Fig 1. Fellow eye with pachydrusen in unilateral polypoidal choroidal vasculopathy.** A 69-year-old female with polypoidal choroidal vasculopathy. (A) Drusen were scattered around the macula on the fundus photography in the right eye and orange-red lesions were located on the macula in the left eye. Late phase indocyanine angiography demonstrated hyperfluorescence corresponding to a white line on color photography in the right eye and polypoidal lesions on macula in the left eye. (B) (Left) A horizontal scan demonstrated a drusenoid deposit(pachydrusen) corresponding to a white line in the right eye. (Right) A horizontal scan though the fovea demonstrated serous retinal detachment with double layer sign.

p = $1.4×10^{-4}$, Group 1: p = 0.001, Group 3: p = 0.019). Gender distribution was not significantly different among the 4 groups, while baseline BCVA was slightly better in Group 1 than Group 2 (p = 0.047). Baseline GLD was slightly larger in Group 2 than Group 1(p = 0.047). Baseline CRT was significantly greater in Group 1 compared with Group 0(p = 0.004) and Group 3 (p = $1.3×10^{-3}$) and baseline GLD was significantly greater in Group 2 than Group 3 (p = 0.029). Baseline subfoveal choroidal thickness (SCT) was greatest in Group 1 among the 4 groups; however, a difference in SCT was seen between Group 1 and Group 2 (p = 0.047).

In all 60 study eyes, mean BCVA significantly improved from 0.44±0.28 at baseline to 0.22 ±0.20 at 6-month, 0.21±0.20 at 12-month, 0.18±0.20 at 18-month, and 0.17±0.20 at 24-month (p = $3.1×10^{-9}$, $2.5×10^{-9}$, $5.7×10^{-10}$ and $9.3×10^{-11}$, respectively). In each group, mean BCVA significantly improved at 24-month (Group 0: p = $1.3×10^{-4}$, Group 1: p = $5.0×10^{-2}$, Group 2: p = $9.5×10^{-6}$, Group 3: p = $4.2×10^{-2}$). Fig 3(A) shows changes of BCVA in each group. After adjusting age, gender, and baseline BCVA, there was not a statistically significant difference in BCVA improvement at 24-month among the 4 groups.

During the 24-month study period, 35(58.3%) eyes required retreatment. The number of retreated eyes was lower in Group 1(12.5%) compared with other groups (p = 0.0038, chi-square test, Group 0: 50%, Group 2: 76.9%, Group 3: 66.7%). Fig 4 shows Kaplan-Meier estimator associated with retreatment-free period among the 4 groups. Retreatment-free period was the longest in Group 1 compared to the other groups(P = 0.0055,Man-Whitney U test) and significant differences in the retreatment-free period were seen between Group 1 and Group 2 (p = 0.0023, log-rank test) and between Group 1 and Group 3 (p = 0.0046, log-rank test). The mean number of additional combination therapy was 0.20, 0, 0.34, 0.33 in Group 0,

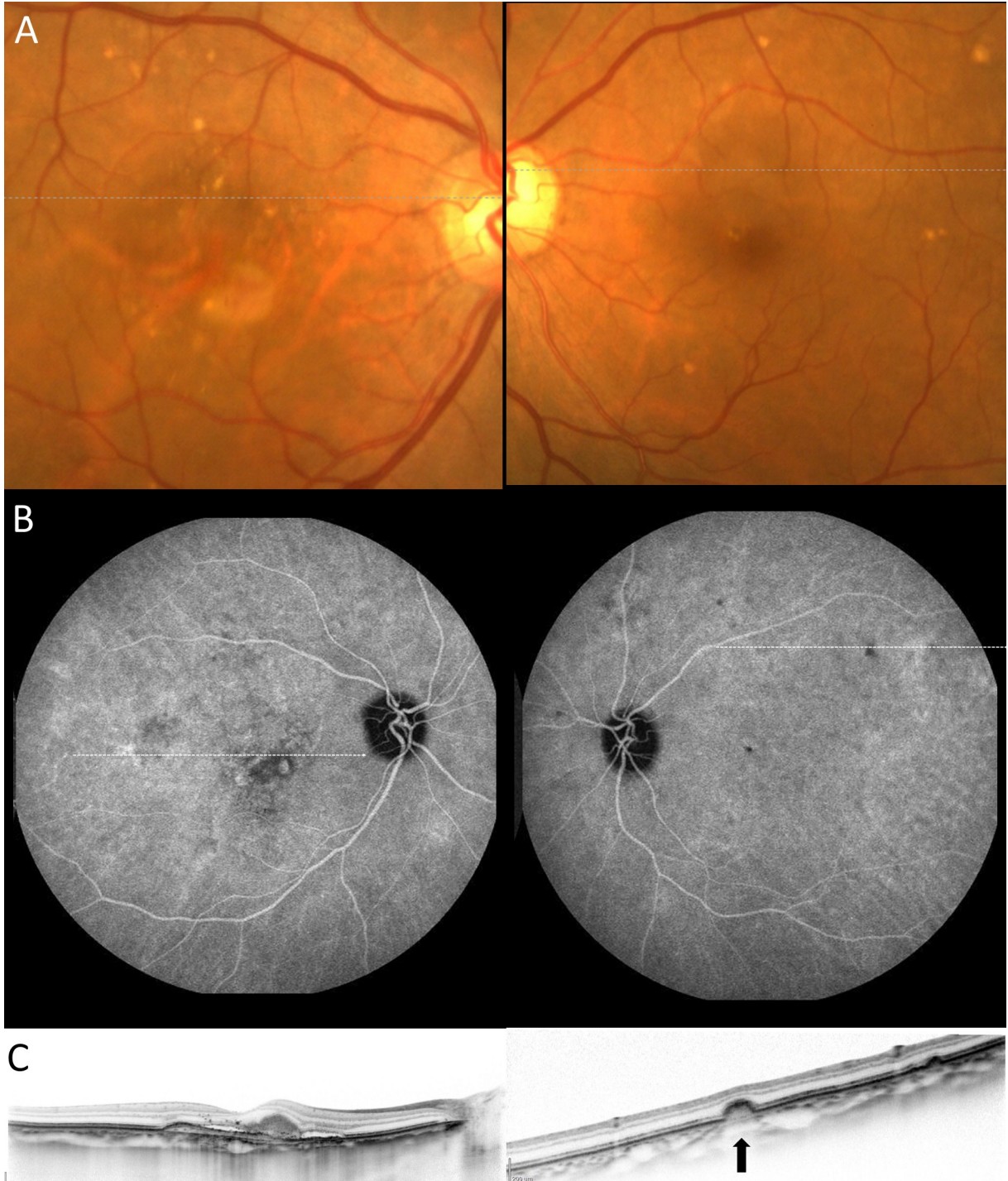

**Fig 2. Fellow eye with soft drusen in unilateral polypoidal choroidal vasculopathy.** A 78-year-old female with polypoidal choroidal vasculopathy. (A) Drusen were scattered around the macula on the fundus photography in both eyes. (B) (Left) The late phase indocyanine green angiography (ICGA) showed that polypoidal lesion on the macula in the right eye. (Right) The late phase angiography revealed hypofluorescent spots corresponding to the drusen. (C) (Left) A horizontal scan through the fovea demonstrated subretinal fluid and subretinal hyperreflective materials on optical coherence tomography (OCT). (Right) A horizontal scan corresponding to a white line on the ICGA showed the bump of retinal pigment epithelium (a black arrow) on OCT.

**Table 1. Demographic characteristics of patients with polyploidal choroidal vasculopathy.**

| | all (n = 60) | Group 0(no drusen) (n = 20) | Group1(pachydrusen) (n = 8) | Group 2 (soft drusen) (n = 26) | Group3 (PCV/scar) (n = 6) |
|---|---|---|---|---|---|
| sex(male) | 41(68.3%) | 12(60%) | 6(75%) | 18(69.2%) | 5(83.3%) |
| age | 72.8±8.4 | 67.9±7.8 | 69.3±4.6 | 78.2±6.9 | 70.5±5.7 |
| logMAR BCVA | 0.44±0.28 | 0.45±0.25 | 0.30±0.11 | 0.52±0.29 | 0.28±0.31 |
| CRT | 382.7±106.0 | 350.3±97.4 | 474.8±72.4 | 399.7±101.5 | 293.8±57.0 |
| SCT | 269.2±94.0 | 269.0±108.5 | 317.6±45.1 | 254.2±86.4 | 270.2±92.4 |
| GLD | 1700.8±763 | 1775±783 | 1288±741 | 1777±733 | 1675±587 |
| CVH on the affected eye | 18(30%) | 10(50%) | 7(87.5%) | 0 | 1(16.7%) |

Log MAR: logarithm of the minimum angle of resolution

CRT: central retinal thickness, SCT: subfoveal choroidal thickness, GLD: greatest linear dimension, CVH: choroidal vascular hyperpermeability

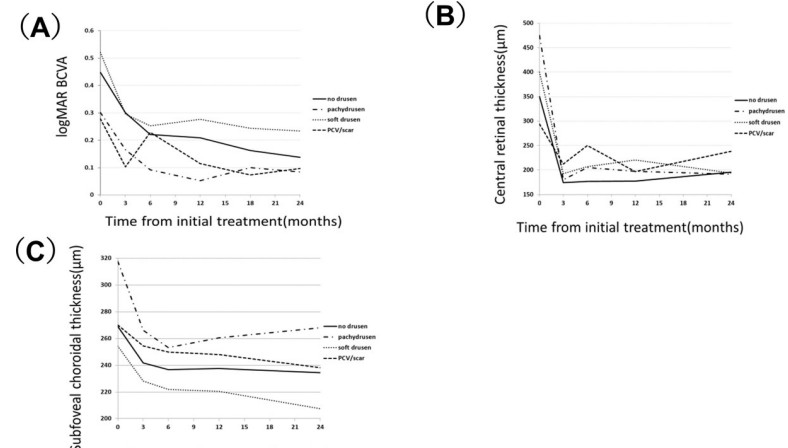

**Fig 3. Changes of best-correct visual acuity, central retinal thickness, and subfoveal choroidal thickness among the 4 groups.** (A) Changes of best-corrected visual acuity among the 4 groups. Baseline visual acuity significantly improved from 0.45±0.26 at baseline to 0.22±0.24, 0.21±0.20, 0.16±0.18 and 0.14±0.17 at 6-month, 12-month, 18-month and 24-month (p = $2.7×10^{-4}$,$1.8×10^{-4}$, $1.8×10^{-4}$, and $1.3×10^{-4}$, respectively) in Group 0. Baseline visual acuity improved from 0.30±0.12 at baseline to 0.09±0.09, 0.05±0.08, 0.10±0.10, 0.08±0.15 at 6-month, 12-month, 18-month and 24-month (p = $1.2×10^{-2}$, $1.2×10^{-2}$, $2.5×10^{-2}$, and $5.0×10^{-2}$, respectively) in Group 1. Baseline visual acuity significantly improved from 0.52±0.29 to 0.25±0.18, 0.28±0.22, 0.24±0.22, 0.23±0.22 at 6-month, 12-month, 18-month and 24-month (p = $1.8×10^{-5}$, $3.1×10^{-4}$, $1.4×10^{-4}$, and $9.5×10^{-6}$, respectively) in Group 2. Baseline visual acuity significantly improved from 0.28±0.34 to 0.23±0.28, 0.12±0.0.14, 0.07±0.20, 0.10±0.23 at 6-month, 12-month, 18-month and 24-month (p = 0.71, 0.14, $2.6×10^{-2}$ and $4.2×10^{-2}$, respectively) in Group 3. (B) Changes of central retinal thickness among the 4 groups. Mean central retinal thickness(CRT) significantly decreased from 350±100 μm at baseline to 177±33μm at 6-month, 178±31μm at 12-month, and 196±62μm at 24-month (p = $8.9×10^{-5}$, $8.9×10^{-5}$ and $2.5×10^{-4}$, respectively) in Group 0, mean CRT also significantly decreased from 475±77 μm at baseline to 205±23μm at 6-month, 197±24μm at 12-month, and 192±33μm at 24-month (p = $1.2×10^{-2}$, $1.2×10^{-2}$ and $1.2×10^{-2}$, respectively) in Group 1, mean CRT significantly decreased from 400±104 μm at baseline to 207±60μm at 6-month, 221±75μm at 12-month, and 194±47μm at 24-month (p = $8.3×10^{-6}$, $3.3×10^{-5}$ and $9.3×10^{-6}$, respectively) in Group 2, and mean CRT decreased from 294±62 μm at baseline to 250±104μm at 6-month, 197±44μm at 12-month, and 238±74μm at 24-month (p = 0.25, $2.8×10^{-2}$ and $5.8×10^{-2}$, respectively) in Group 3. There were no significant differences in CRT at 24-month among the 4 groups (p = 0.32, analysis of variance). (C) Changes of subfoveal choroidal thickness among the 4 groups. Mean subfoveal choroidal thickness(SCT) significantly decreased from 269±111 μm at baseline to 237 ±101 μm at 6-month, 238±104 μm at 12-month, and 234±98 μm at 24-month (p = $2.5×10^{-4}$, $7.2×10^{-4}$ and $1.2×10^{-4}$, respectively) in Group 0, mean SCT also significantly decreased from 318±48 μm at baseline to 253±36 μm at 6-month, 260±53 μm at 12-month, and 268±40 μm at 24-month (p = $1.7×10^{-2}$, $1.2×10^{-2}$ and $1.2×10^{-2}$, respectively) in Group 1, mean SCT significantly decreased from 254±88 μm at baseline to 222±83 μm at 6-month, 220±87 μm at 12-month, and 208±81 μm at 24-month (p = $3.5×10^{-5}$, $9.9×10^{-6}$ and $8.3×10^{-6}$, respectively) in Group 2, and mean SCT decreased from 270±101 μm at baseline to 250±101 μm at 6-month, 248±92 μm at 12-month, and 238±92 μm at 24-month (p = $2.8×10^{-2}$, $8.0×10^{-2}$ and $2.5×10^{-2}$, respectively) in Group 3. There were no significant differences in SCT at 24-month among the 4 groups (p = 0.32, analysis of variance).

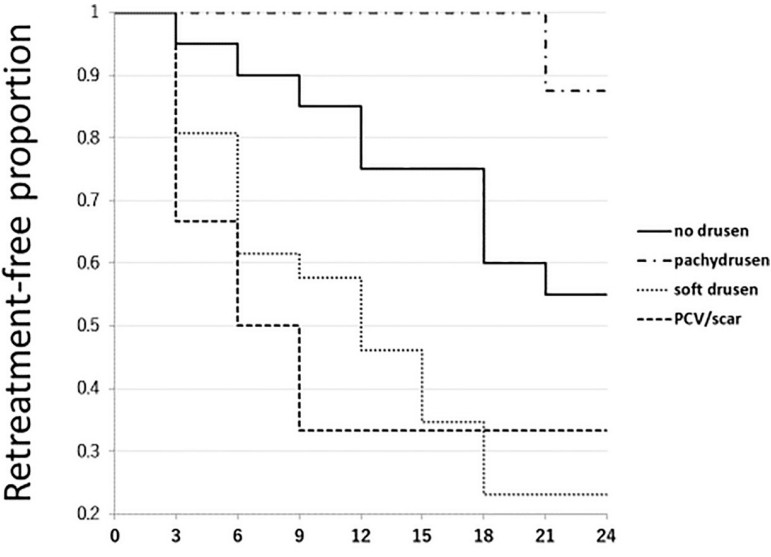

**Fig 4. Kaplan-Meier estimator showing retreatment-free period after the initial combination therapy.** Mean retreatment-free period after the combination therapy was 19.1±6.9 (95%CI:15.8–22.3), 23.6±1.1 (95%CI:22.7–24.0), 12.8±7.9 (95%CI:9.6–16.0), 11.5±9.9 (95% CI:1.1–21.9) in Group 0, Group 1, Group 2, Group 3, respectively. Although it did not reach a statistical difference in retreatment-free period between Group 0 and Group 1(p = 0.072, log-rank test), there were significant differences in retreatment-free period between Group 1 and Group 2 (p = 0.0023,log-rank test), and between Group 1 and Group 3 (p = 0.0046, log-rank test).

Group 1, Group 2, and Group 3, respectively, which was not significantly different among the 4 groups (p = 0.43, analysis of variance). Mean number of additional intravitreal injections of aflibercept was 0.85±1.35, 0.125±0.35, 2.46±2.58, 2.33±2.34 in Group 0, Group 1, Group 2, and Group 3, respectively. Eyes in Group 1 required less additional IAI compared to other groups (P = 0.017, chi-square test). A statistically significant difference was seen between Group 1 and Group 2 (p = 0.009, Mann-Whitney U test) and between Group 0 and Group 2 (p = 0.015, Mann-Whitney U test).

In all 60 study eyes, mean CRT significantly decreased from 269±94 μm at baseline to 234 ±86μm at 6-month, 234±89μm at 12-months, and 228±85μm at 24-months (p = $3.4\times10^{-11}$, $7.6\times10^{-11}$ and $4.8\times10^{-11}$, respectively) and Fig 3(B) shows changes of central retinal thickness among the 4 groups. Fig 3(C) shows changes of subfoveal choroidal thickness among the 4 groups.

## Discussion

Drusen are extracellular materials located between Bruch's membrane and the retinal pigment epithelium. Although they are a manifestation of the normal aging process when their size is less than 63 μm, they are generally considered early signs of AMD when they exceed 63 μm in size. The risk for progression to advanced AMD varies depending on size, types and distribution pattern of drusen, and fellow eye condition. [26–29] Therefore, it is important for physicians to differentiate drusen types and understand characteristics of each drusen type. However, there have been a few studies in the literature examining treatment outcomes in eyes with pachydrusen as they are a relatively recently described clinical entity. Our study demonstrated that among eyes with PCV, fellow eyes with pachydrusen were predictive of decreased recurrence and treatment burden.

In the present study, we classified patients with PCV into 4 groups (Group 0: no drusen, Group 1: pachydrusen, Group 2; soft drusen, Group 3; PCV/ fibrovascular scarring) based upon the untreated fellow-eye and investigated whether there are significant differences in response to the combination therapy for PCV among the 4 groups. Mean retreatment-free period was significantly longer in Group 1(fellow eye with pachydrusen) (Group 0: 19.1±6.9, Group 1: 23.6±1.1, Group 2: 12.8±7.9, Group 3: 11.5±9.9 months, p = 0.001, analysis of variance). Therefore, the additional number of retreatments was also decreased in Group 1.

There are several reasons to explain the present results. Firstly, it has been reported that eyes with thickened subfoveal choroidal thickness lead to a favorable course including visual improvement and chance of recurrence after the combination therapy for PCV. [30] In the present study, subfoveal choroidal thickness was significantly thicker in Group 1 compared to the other 4 groups, a finding which was consistent with previous reports. [21–23] Second, it has been reported that choroidal vascular hyperpermeability (CVH) is associated with the need for less additional anti-VEGF injections at 12 months among PCV eyes receiving combination PDT and anti-VEGF therapy. [31] CVH was first reported in eyes with CSC as patchy area of ICGA hyperfluorescence seen best in mid- and late-phase study. [32] CVH has been considered angiographic evidence of a disturbance of choroidal vascular circulation. It has been recently reported that pachyvessels traversed the area of CVH on mid and late phase ICGA, suggesting that a thickened outer choroid is associated with development of CVH. [33] In the present study, CVH was present in18 eyes (30%), which is similar to the figure in previous reports [31, 34] and CVH is most frequently seen in Group 1. Third, several studies have reported that older age is a risk factor for recurrence/retreatment in treatment for exudative AMD, including anti-VEGF monotherapy or the combination therapy. [35–37] Eyes from Group 1 eyes are youngest among the cohort. In a recent report, exudative AMD patients with pachydrusen were younger than those with soft drusen or pseudodrusen. Pachydrusen might be early onset compared with soft drusen and pseudodrusen. At 2 years, mean BCVA significantly improved from baseline (0.44±0.28) to their final visit at month 24 (0.17±0.20) in all 60 study eyes (p = $9.3×10^{-11}$). As reported previously, [10, 14, 37] combination therapy of PDT and IAI results in favorable outcomes at 2 years. Although we previously reported that visual improvement differs depending on fellow eye condition when treating with intravitreal aflibercept monotherapy for neovascular AMD, [38] there were not any significant differences in visual improvement among the 4 groups in this study.

The pachychoroid phenotype was characterized by the presence of a pachyvessel, focal or diffuse choroidal thickening on OCT and CVH on ICGA. Further, the absence or paucity of soft drusen is also a feature of pachychoroid. Of note none of the study eyes with soft drusen had CVH, further supporting the distinction of the pachychoroid phenotype. Most eyes with pachydrusen also have pachychoroid features including CVH as is in the case in the present study. Considering the close link between pachychoroid and pachydrusen, authors feel that eyes with pachydrusen and PCV lend credence to the idea that a choroidal mechanism is mediating the exudation. PDT is effective for pachychoroid diseases including CSC and PCV with pachychoroid features by normalizing the thickening choroid; on the other hand, eyes with soft drusen and exudation are considered drusen mediated exudation; Soft drusen may also induce inflammation in the retina, which explain why eyes with soft drusen appear to have increased recurrence of exudation due to the theory of an inflammatory mediated mechanism of exudation. Thus, the authors feel that two pathways may lead to exudation, a choroidal mediated and a drusen mediated one. These differing mechanisms for exudation may explain the different rates of recurrence.

Our study demonstrates the importance of assessing the clinical characteristics among untreated fellow eyes in treatment naïve PCV patients as fellow eye characteristics are predictors of recurrence, recurrence-free period, and number of additional retreatments.

There were several limitations in this study. The major limitation of the present study is a retrospective nature of analysis and small sample size, especially Group1 and Group 3. A large-scale prospective study would be needed to confirm this tentative conclusion. To the best of our knowledge, this is the first report that demonstrated patients with pachydrusen in fellow eyes were good responders to the combination therapy and may have a lower treatment burden for PCV. Further study is needed to determine whether patients with pachydrusen also have similar responses to other treatment modalities including anti-VEGF monotherapy.

In summary, untreated fellow eye characteristics may predict recurrence, the recurrence-free period after the combination therapy for PCV and subsequent treatment burden.

## Supporting information

**S1 Data.**
(XLS)

## Author Contributions

**Conceptualization:** Yoichi Sakurada.

**Data curation:** Mio Matsubara, Yoichi Sakurada, Atsushi Sugiyama, Yoshiko Fukuda.

**Writing – original draft:** Mio Matsubara, Yoichi Sakurada.

**Writing – review & editing:** Atsushi Sugiyama, Yoshiko Fukuda, Ravi Parikh, Kenji Kashiwagi.

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
