## [Decision Letter · Decision Letter 0]

5 Mar 2020

PONE-D-20-00729

Response to photodynamic therapy combined with intravitreal aflibercept for polypoidal choroidal vasculopathy depending on fellow-eye condition:2-year results

PLOS ONE

Dear Dr Sakurada,

Thank you for submitting your manuscript to PLOS ONE. After careful consideration, we feel that it has merit but does not fully meet PLOS ONE’s publication criteria as it currently stands. Therefore, we invite you to submit a revised version of the manuscript that addresses the points raised during the review process.

We would appreciate receiving your revised manuscript by Apr 19 2020 11:59PM. To enhance the reproducibility of your results, we recommend that if applicable you deposit your laboratory protocols in protocols.io, where a protocol can be assigned its own identifier (DOI) such that it can be cited independently in the future. For instructions see: http://journals.plos.org/plosone/s/submission-guidelines#loc-laboratory-protocols

We look forward to receiving your revised manuscript.

Kind regards,

Manuel Alberto de Almeida e Sousa Falcão, M.D., Ph.D.

Academic Editor

PLOS ONE

Journal Requirements:

2. In the Methods section and the online submission form, please provide additional information about the patient records used in your retrospective study. Specifically, please ensure that you have discussed whether all data were fully anonymized before you accessed them and/or whether the IRB or ethics committee waived the requirement for informed consent. If patients provided informed written consent to have data from their medical records used in research, please include this information.

4. Your ethics statement must appear in the Methods section of your manuscript. If your ethics statement is written in any section besides the Methods, please move it to the Methods section and delete it from any other section. Please also ensure that your ethics statement is included in your manuscript, as the ethics section of your online submission will not be published alongside your manuscript.

Reviewers' comments:

Reviewer's Responses to Questions

**Comments to the Author**

1. Is the manuscript technically sound, and do the data support the conclusions?

Reviewer #1: No

Reviewer #2: Partly

2. Has the statistical analysis been performed appropriately and rigorously? 

Reviewer #1: No

Reviewer #2: Yes

3. Have the authors made all data underlying the findings in their manuscript fully available?

Reviewer #1: Yes

Reviewer #2: Yes

4. Is the manuscript presented in an intelligible fashion and written in standard English?

Reviewer #1: No

Reviewer #2: Yes

5. Review Comments to the Author

Reviewer #1: I read with interest the present paper which explores the differences between treatment response in PCV patients according to the type of drusen found in the fellow eye. It is a original ideia and interesting to explore in the search of biomarkers of disease prognosis and response to treatment.

However, there are several issues that must be addressed before accepting the manuscript for publication.

First, the short title has a minor error in "depending"

Introduction:

The authors start by stating that "PCV is a unique subtype of exudative age-related macular degeneration (AMD) ", this, however, is still a debated statement. Controversy persists, especially as PCV can be found in other settings and pathologic associations.

The introduction is somewhat globally inconsistent and better rephrasing and English review is necessary before publication.

Combination treatment versus anti-VEGF monotherapy approach is still not consensual and is not the point of the present study, but assumptions of superiority of one over the other cannot be made as expressed in the second paragraph. Other bibliography sources are to be added to support the information provided if this point is presented here.

Pachydrusen are presented as a distinct clinical entity, but reference and linkage to pachychoroid spectrum should be more clearly explored. Should these pachydrusen be completely assumed to be part of AMD, or they can be present in other settings? The group refers to a previous study on clinical and genetic characteristics of pachydrusen in patients with exudative AMD, where they acknowledge that patients with pachydrusen have genetic and clinical characteristics distinct from those of soft drusen and pseudodrusen of typical AMD, however the presence of both pachydrusen, pachychoroid and PCV in the setting of other diseases is not further explored.

Methods:

Again, English is to be improved, for example in the line 85 " slit-lamp examination with or without 78 diopter lens intraocular measurement".

Only one vertical and one horizontal line in OCT analysis are enough? How was the CRT values obtained? And choroidal thickness?

Only late-phase ICGA was used to discriminate between soft and pachydrusen but other features must be considered, only one reference for this ICGA based definition is not sufficient.

Since PCV is stated here to be considered as a variant of neovascular AMD, was monthly loading dose performed or only one combination treatment was performed with observation after 3 months? As this is not standard for PCV treatment or typical nAMD treatment, was follow-up of only once every 3 months considered sufficient?

Was the t-test possible to use in this sample? Groups are quite small - group 1 has only 8 subjects and is the main focus of this work.

Results:

Line 124: The group's enumeration in methods is wrong: 1, 2, 3, and 3 again. Different from the Results - Table: from 0 to 4, the results are therefore not understandable in the subsequent sections when comparisons between groups are made.

Group 0 - CNV with no drusen in the fellow eye. Is this group to be considered AMD? Or is CNV due to other causes? Such asymmetry between eyes would not be expected. Can these 4 groups be truly comparable? It seems the authors are mixing different causes of PCV, and therefore different results are expected regarding treatment.

Hyperpermeability of the choroid is substantially superior in the group with pachydrusen - again pathophysiology should be further explored in the discussion, before discussing different responses to treatment.

Discussion:

The work is much interesting from a clinical perspective, as biomarkers of treatment and prognostic are necessary to improve outcomes in CNV treatment, but the authors should investigate and elaborate more on the difference of phenotypes and if these could represent in fact variants of the same disease process or distinct clinical entities with PCV as the final result. Comparing treatment outcomes is not sufficient per se. Instead of only focusing in pachydrusen, the complete pathophysiologic picture and pachychoroid spectrum versus typical AMD and then response to treatment should be considered and discussed.

Reviewer #2: Dear Authors,

The paper reports some interesting results in a very hot topic in AMD treatment, the underestimated PCV. The authors presents some findings that are very important to observe regarding the fellow eye conditions and how the affected eye responds to the proposed therapy. The combined therapy PDT + aflibercept for PCV appears to be a good choice at a very interesting rationale, however the availability of verteporfin in some countries is limited. There is a trend for anti-VEGF mono therapy for most of those case, and some papers shows very similar results.

Regarding the figures, the authors illustrate two examples of patients in group 2 and 3, showing the differences of pachydrusen and soft drusen. The other figures illustrate the main results as described in the methods.

It is important to remember that this is a retrospective study, with uneven number of patients on each group, specially in group 1 and 3, with only 8 and 6 patient respectively, almost 1/3 of the patients of the other groups, this might affect the comparative results. The most interesting of the paper was to observe that patients with pachydrusen in the FE had the longest re-treatment free period and less additional IV injections, that might reflect a better and effective response to PDT as pointed to the authors.

Overall the paper is well written, some english corrections should be made, and in my opinion the authors should enrich their discussion. Of course is not possible to make definitive conclusion with a retrospective studies but may suggest some insights for future papers.

All the best

6. PLOS authors have the option to publish the peer review history of their article (what does this mean?). If published, this will include your full peer review and any attached files.

Reviewer #1: No

Reviewer #2: No

---

## [Author Response · Author response to Decision Letter 0]

11 May 2020

Dear Academic Editor: 

We thank you, and the reviewers for timely and constructive feedback on our manuscript. We appreciate the opportunity to respond.

Reviewer #1: I read with interest the present paper which explores the differences between treatment response in PCV patients according to the type of drusen found in the fellow eye. It is a original ideia and interesting to explore in the search of biomarkers of disease prognosis and response to treatment.

However, there are several issues that must be addressed before accepting the manuscript for publication.

Reply: Thank you for your timely and positive feedback on our manuscript.

First, the short title has a minor error in "depending"

Reply: Thank you for your pointing. We corrected the term from “depnding” to “depending”.

Introduction:

The authors start by stating that "PCV is a unique subtype of exudative age-related macular degeneration (AMD) ", this, however, is still a debated statement. Controversy persists, especially as PCV can be found in other settings and pathologic associations. The introduction is somewhat globally inconsistent and better rephrasing and English review is necessary before publication.

Reply: We agree and have emphasized that although some feel PCV is a subtype of AMD, there is debate as to whether PCV is a distinct entity. Please see additions to line 44, 50-51 to reflect these changes. The manuscript was reviewed and edited by one (R.P) of the authors, a native speaking US retinal specialist with prior publications and research experience to ensure improved clarity and phrasing in English.

Combination treatment versus anti-VEGF monotherapy approach is still not consensual and is not the point of the present study, but assumptions of superiority of one over the other cannot be made as expressed in the second paragraph. Other bibliography sources are to be added to support the information provided if this point is presented here.

Reply: We agree and have added citations from clinical trials such as EVEREST 1 and 2 demonstrating superiority of both anatomic and visual outcomes of combination therapy over PRN anti-vegf monotherapy.

Pachydrusen are presented as a distinct clinical entity, but reference and linkage to pachychoroid spectrum should be more clearly explored. Should these pachydrusen be completely assumed to be part of AMD, or they can be present in other settings? The group refers to a previous study on clinical and genetic characteristics of pachydrusen in patients with exudative AMD, where they acknowledge that patients with pachydrusen have genetic and clinical characteristics distinct from those of soft drusen and pseudodrusen of typical AMD, however the presence of both pachydrusen, pachychoroid and PCV in the setting of other diseases is not further explored.

Reply: Thank you for your comments. Pachydrusen are often seen in eyes with pachychoroid diseases including central serous chorioretinopathy, pachychoroid neovasculopathy, and polypoidal choroidal vasculopathy. Therefore, these manifestations are not specific to AMD. We added the following sentences“ Pachydrusen are not specific to AMD and they are often seen in pachychoroid diseases such as central serous chorioretinopathy”.

Methods:

Again, English is to be improved, for example in the line 85 " slit-lamp examination with or without 78 diopter lens intraocular measurement".

Reply: Thank you for your pointing. The term “intraocular measurement” was removed. The manuscript was reviewed and edited by one (R.P) of the authors (U.S retinal specialist).

Only one vertical and one horizontal line in OCT analysis are enough? How was the CRT values obtained? And choroidal thickness?

Reply: We are happy to clarify these things. Please see lines 103-108. We performed a horizontal and vertical OCT scan through the fovea. In patients with extrafoveal lesions, we performed OCT scans corresponding to the lesion in addition to scans of the fovea. This scan pattern might be insufficient; however, we cannot judge whether this scan pattern has influence on the present results. 

CRT was measured as the vertical distance between inner surface of neurosensory retina and retinal pigment epithelium beneath the fovea. Subfoveal choroidal thickness was measured as the vertical distance between Bruch’s membrane and choroidoscleral border at the fovea. 

Only late-phase ICGA was used to discriminate between soft and pachydrusen but other features must be considered, only one reference for this ICGA based definition is not sufficient.

Reply: Please see lines 95-97 and we have added how we defined soft drusen and pachydrusen on OCT imaging as well. OCT scans demonstrate that soft drusen are RPE bump while pachydrusen are drusenoid deposits. In this retrospective study, we did not perform volume-scan OCT covering posterior pole for all study eyes. Therefore, it is difficult to differentiate all drusen in the posterior pole. We cited only reference [24]; however, a previous report demonstrated that soft drusen exhibited hypofluorescent. (Arnold JJ et al. Indocyanine green angiography of drusen. Am J Ophthalmol. 1997). We cited this article as reference [25].

Since PCV is stated here to be considered as a variant of neovascular AMD, was monthly loading dose performed or only one combination treatment was performed with observation after 3 months? As this is not standard for PCV treatment or typical nAMD treatment, was follow-up of only once every 3 months considered sufficient?

Reply: In Japan and other nations outside of the US, often PRN treatment is necessary as monthly dosing is not always feasible. In our study, an initial anti-VEGF was given to address acute exudation and PDT was done for longer term therapy of aneurysmal lesions. Patients were followed every 3-month intervals until recurrence. This approach involving PDT and anti-VEGF agents was first reported by Sato et al in 2010 (Am J Ophthalmol). In this report, a follow-up interval was 3-month was used and our current standard of care. Following their report, we determined the follow-up period until appearance of recurrent exudation. We considered that our follow-up period (3-month) was adequate until recurrence based on prior studies and treatment patterns in Japan.

Was the t-test possible to use in this sample? Groups are quite small - group 1 has only 8 subjects and is the main focus of this work.

Reply: Instead of a t-test, differences of values between before and after treatment were analyzed by Wilcoxon signed rank test. Therefore, all p-values were tested by Wilcoxon signed rank test was changed.

Results:

Line 124: The group's enumeration in methods is wrong: 1, 2, 3, and 3 again.

Reply: Thank you for bringing this to our attention. We corrected the numbering from Group 0 to Group 3.

Different from the Results - Table: from 0 to 4, the results are therefore not understandable in the subsequent sections when comparisons between groups are made.

Reply: We apologize for any confusions. The groups are label 0-3 for a total of 4 groups. The tables are correct to the best of our knowledge. 

Group 0 - CNV with no drusen in the fellow eye. Is this group to be considered AMD? Or is CNV due to other causes? Such asymmetry between eyes would not be expected. Can these 4 groups be truly comparable? It seems the authors are mixing different causes of PCV, and therefore different results are expected regarding treatment.

Reply: Thank you for your comments. We wish to clarify that minimal or a complete lack of drusen are typically the case in PCV as well as other pachychoroid phenotypes such as CSC (Maruko et al, Am J Ophthalmol, 2007). Further, we also wish to highlight in the text both proposed mechanisms of exudation in neovascular retinal disease which are choroid mediated (ie pachychoroid) or inflammatory (ie from drusen). 

Please see lines 341-358 which discusses phenotypical differences in PCV and traditional AMD as well as proposed mechanisms of exudation.

.

Hyperpermeability of the choroid is substantially superior in the group with pachydrusen - again pathophysiology should be further explored in the discussion, before discussing different responses to treatment.

Replay: Thank you for valuable comments. We agree and will add a discussion on the above. 

Please see lines 319-325 including the added citations [32] and [33].

Discussion:

The work is much interesting from a clinical perspective, as biomarkers of treatment and prognostic are necessary to improve outcomes in CNV treatment, but the authors should investigate and elaborate more on the difference of phenotypes and if these could represent in fact variants of the same disease process or distinct clinical entities with PCV as the final result. Comparing treatment outcomes is not sufficient per se. Instead of only focusing in pachydrusen, the complete pathophysiologic picture and pachychoroid spectrum versus typical AMD and then response to treatment should be considered and discussed.

Reply: Thank you for your thoughtful comments. We agree and have added the above. Please see lines 341-358.

Reviewer #2: Dear Authors,

The paper reports some interesting results in a very hot topic in AMD treatment, the underestimated PCV. The authors present some findings that are very important to observe regarding the fellow eye conditions and how the affected eye responds to the proposed therapy. The combined therapy PDT + aflibercept for PCV appears to be a good choice at a very interesting rationale, however the availability of verteporfin in some countries is limited. There is a trend for anti-VEGF mono therapy for most of those case, and some papers shows very similar results.

Regarding the figures, the authors illustrate two examples of patients in group 2 and 3, showing the differences of pachydrusen and soft drusen. The other figures illustrate the main results as described in the methods.

It is important to remember that this is a retrospective study, with uneven number of patients on each group, especially in group 1 and 3, with only 8 and 6 patient respectively, almost 1/3 of the patients of the other groups, this might affect the comparative results. The most interesting of the paper was to observe that patients with pachydrusen in the FE had the longest re-treatment free period and less additional IV injections, that might reflect a better and effective response to PDT as pointed to the authors.

Overall the paper is well written, some English corrections should be made, and in my opinion the authors should enrich their discussion. Of course, is not possible to make definitive conclusion with retrospective studies but may suggest some insights for future papers.

Reply: Thank you for your positive comments and constructive criticism. 

1. PDT is still commonly performed for eyes with PCV in Asia despite a complicated procedure. Due to costs, in most of Asia anti-VEGF is PRN and thus PDT is often used to reduce treatment burden as it is not practical for most patients to receive continuous anti-VEGF if no active exudation. Clinical trial data such as the EVEREST II and FUJISAN studies have shown the decreased treatment burden of combination therapy compared to anti-VEGGF monotherapy. 

2. As the reviewer pointed, we acknowledged that weak point of the current study is retrospective nature of analysis and small number of Group 1 and Group 3. We added the several sentences to improve the manuscript in the discussion.

3. The manuscript has been edited by one of the authors (R.P) a retinal specialist in US.

Thank you again for reviewing our manuscript.

---

## [Decision Letter · Decision Letter 1]

24 Jul 2020

Response to photodynamic therapy combined with intravitreal aflibercept for polypoidal choroidal vasculopathy depending on fellow-eye condition:2-year results

PONE-D-20-00729R1

Dear Dr. Sakurada,

We’re pleased to inform you that your manuscript has been judged scientifically suitable for publication and will be formally accepted for publication once it meets all outstanding technical requirements.

Kind regards,

Manuel Alberto de Almeida e Sousa Falcão, M.D., Ph.D.

Academic Editor

PLOS ONE

Additional Editor Comments (optional):

Reviewers' comments:

Reviewer's Responses to Questions

**Comments to the Author**

1. If the authors have adequately addressed your comments raised in a previous round of review and you feel that this manuscript is now acceptable for publication, you may indicate that here to bypass the “Comments to the Author” section, enter your conflict of interest statement in the “Confidential to Editor” section, and submit your "Accept" recommendation.

Reviewer #1: All comments have been addressed

Reviewer #2: All comments have been addressed

2. Is the manuscript technically sound, and do the data support the conclusions?

Reviewer #1: Partly

Reviewer #2: Yes

3. Has the statistical analysis been performed appropriately and rigorously? 

Reviewer #1: Yes

Reviewer #2: Yes

4. Have the authors made all data underlying the findings in their manuscript fully available?

Reviewer #1: Yes

Reviewer #2: Yes

5. Is the manuscript presented in an intelligible fashion and written in standard English?

Reviewer #1: Yes

Reviewer #2: Yes

6. Review Comments to the Author

Reviewer #1: (No Response)

Reviewer #2: The authors made required changes in the final paper and all comments that were made was addressed by the authors

7. PLOS authors have the option to publish the peer review history of their article (what does this mean?). If published, this will include your full peer review and any attached files.

Reviewer #1: No

Reviewer #2: **Yes: **Fernando Marcondes Penha